# Migration Route of *Sthenoteuthis oualaniensis* in the South China Sea Based on Statolith Trace Element Information

**DOI:** 10.3390/ani13182811

**Published:** 2023-09-05

**Authors:** Jiangtao Fan, Zhou Fang, Shengwei Ma, Peng Zhang, Xue Feng, Zuozhi Chen

**Affiliations:** 1South China Sea Fisheries Research Institute, Guangzhou 510300, China; fjt@scsfri.ac.cn (J.F.); msw@scsfri.ac.cn (S.M.); zhangpeng@scsfri.ac.cn (P.Z.); fengx122@163.com (X.F.); 2Key Laboratory for Sustainable Utilization of Open-Sea Fishery, Ministry of Agriculture and Rural Affairs, Guangzhou 510300, China; 3College of Marine Sciences, Shanghai Ocean University, Shanghai 201306, China; zfang@shou.edu.cn; 4Southern Marine Science and Engineering Guangdong Laboratory (Guangzhou), Guangzhou 511458, China

**Keywords:** South China Sea, *Sthenoteuthis oualaniensis*, statolith, trace element, migration route

## Abstract

**Simple Summary:**

*Sthenoteuthis oualaniensis* is an economically important cephalopod in the South China Sea. This species demonstrates rapid growth and has a complex population structure and a wide range of migration. It is an important predator and prey in its ecosystem. Statoliths are hard cephalopod tissues, and the biogeochemical information they contain is an effective material for analyzing the characteristics of the organism’s life history. The migration route of *S. oualaniensis* is not clear. Using a ratio of trace elements to predict and calculate a range of potential habitat sea areas, this study found that the winter stock of *S. oualaniensis* hatched in the southern South China Sea, and the larvae then migrated northwest during the summer monsoon. The summer–autumn stocks hatched in the northern South China Sea, and the larvae migrated southward under mesoscale closed, anticyclonic circulation in the northern South China Sea. These results provide insight into the migration of *S. oualaniensis* in the South China Sea.

**Abstract:**

*Sthenoteuthis oualaniensis* (Lesson, 1830) is a pelagic species with a complex population structure and wide migration range. The trace elements in statoliths are effective indicators for reconstructing the life history of an individual. In this study, the trace elements in statoliths were determined via laser ablation inductively coupled plasma mass spectrometry, and a multiple regression tree (MRT) model was used to trace the migration of *S. oualaniensis* and identify its potential habitats in the South China Sea. Na, Mg, Fe, Sr, and Ba were the effective trace elements, with significant differences found among stocks (*p* < 0.05). The MRT was divided into five clusters representing five life history stages. The Mg:Ca and Sr:Ca ratios decreased initially and increased thereafter, and the Mg:Ca, Sr:Ca, and Ba:Ca ratios differed significantly among the stages of the life history in each stock (*p* < 0.05). The hatching water temperatures for the winter and summer–autumn spawning populations were 28.05–28.88 °C (temperature at 25 m) and 27.15–27.92 °C (temperature at 25 m). The winter stock hatched in the southern South China Sea, and the larvae then migrated northwest during the summer monsoon. The summer–autumn stocks hatched in the northern South China Sea, and the larvae migrated southward under the mesoscale closed anticyclonic circulation in the northern South China Sea. These results provide insight into the migration of *S. oualaniensis* in the South China Sea.

## 1. Introduction

Cephalopods are important not only as predators but also as prey in the oceanic ecosystem [1,2]. Ommastrephidae species are mainly distributed in open waters and are characterized by a short life cycle, multiple-season stocks, and long-distance migration [3,4]. Understanding migration in these species, including the biotic and abiotic determinants and spatiotemporal patterns, will help us understand their population dynamics and key habitat needs at different life history stages [5,6,7,8]. The microchemical information contained in biomineralized marine tissues is a natural label that can reveal individual habitat histories and reflect changes in the marine environment [6,7,8,9,10].

Statoliths are pairs of hard tissues located in the balance sac; they have a stable external shape and a clear and regular growth pattern that continues throughout the entire life history of the organism [11,12,13]. Trace elements deposited on statoliths are important indicators of environmental conditions [14]. Liu et al. used the ratios of trace elements in statoliths, Mg:Ca and Sr:Ca, to determine the population structure of *Dosidicus gigas* in waters off the coast of Chile and Peru and reconstructed the habitat of the population [5]. Zumholz et al. speculated that adult *Gonatus fabricii* migrated to cold water areas for feeding based on changes in the ratios of U/Ca and Sr/Ca [15]. In addition, the migratory route of *Uroteuthis chinensis* in the South China Sea was determined based on Sr/Ca and Ba/Ca values, revealing differences in the element ratios of individuals at different life history stages [16]. Therefore, analyses of the composition of trace elements in species’ statoliths and its relationship with environmental factors can provide insight into the life history of individuals.

Some of the trace elements in a statolith come from nutrients in the embryonic stage, and the rest come from the absorption and combination of the anions and cations in seawater [4]. During this process, the elemental concentrations in the statolith are affected by their properties and affinities and chemical reactions during the transport process and can reflect the aquatic environment of the cephalopod throughout its life cycle [4,17,18]. Therefore, trace elements are considered good carriers of information for revealing the history of an individual and effective indicators of changes in the marine environment; they are widely used in the reconstruction of cephalopod habitats [5,14,19].

*S. oualaniensis* is a shallow sea species that is widely distributed around the world, particularly in the western Pacific Ocean and the South China Sea [20,21,22]. In the periods 2003–2005, 2013–2014, and 2015–2016, China carried out exploratory fishing operations to identify squid resources in the northern Indian Ocean, the central and western Pacific, and equatorial waters and found a stable reserve fishing ground with abundant kite squid resources for sustainable development and utilization [3,23,24]. A large number of potential squid sites have also been found in the South China Sea in recent years. They are mainly composed of medium-form and dwarf-form stocks characterized by light-emitting organs on the back [3,25,26]. In the South China Sea ecosystem, squids mainly feed on crustaceans, small pelagic fishes, and small cephalopods and are an important food source for large predators [26,27].

In the South China Sea, *S. oualaniensis* can be divided into four seasonal stocks based on hatching dates [3,22]. Usually, squid caught in the spring are mainly from the summer stock (hatched in the June–August period), squid caught in the summer are mainly from the autumn stock (hatched in the September–November period), squid caught in the autumn are mainly from the winter stock (hatched in December and January), and squid caught in the winter are mainly from the spring stock (hatched in March and May) [2]. Within 160–260 days, the growth rates of the mantle length and body weight are highest in the autumn stock [3]. In order to meet their feeding and reproductive needs, different stocks practice reproduction and feeding migration [22,23]. In addition, variations in habitat can result in differences in growth patterns and migration paths among stocks [3,22,28]. Therefore, to reveal the life history characteristics of *S. oualaniensis*, it is necessary to study the migration routes of different groups and habitat preferences at different stages.

Therefore, in this study, laser ablation inductively coupled plasma mass spectrometry (LA-ICP-MS) was used to determine the trace elements in *S. oualaniensis* statoliths for comparisons of the ratios of trace elements between growth stages and analyses of differences in life history among stocks. According to the optimal model for the relationship between the water temperature and the trace elements in statoliths, the range of potential habitats and the migration routes of different stocks were predicted and reconstructed, improving our understanding of the migration of *S. oualaniensis* in the South China Sea.

## 2. Materials and Methods

### 2.1. Survey Time and Sea Area

The survey was performed in March, May, and September in 2020 in the South China Sea (110° E to 120° E, 10° N to 22° N) (Figure 1). The survey vessel was the “Nan Feng” fishery resource survey vessel, and some of parameters of this vessel are as follows: a total length of 66.66 m, a total width of 12.4 m, a gross tonnage of 1980 tons, a cabin capacity of 358 cubic meters, and the power of the main engine was 1920 kW.

### 2.2. Environmental Data

Seawater temperature affects biological characteristics such as individual growth, reproduction, and migration [3]. The temperatures at different water layers affect population distribution and habitats [3,29]. In this study, the temperature was evaluated at different water layers, including depths of 5 m (Temp_5), 25 m (Temp_25), 55 m (Temp_55), 75 m (Temp_75), and 105 m (Temp_105). Temperature data were downloaded from the website of the US National Oceanic and Atmospheric Administration (http://apdrc.soest.hawaii.edu/las/v6, accessed on 3 March 2023). The spatial resolution of all the downloaded temperature variables was processed via MatLab 7.0 to be 0.25° × 0.25°.

### 2.3. Methods

#### 2.3.1. Statolith Measurements

The mantle length (ML) (accurate to 0.1 cm), body weight (BW) (accurate to 0.1 g), sex, and gonad maturity grade were measured (Table 1) [3]. Each statolith was removed from the head and stored in a 1.5 mL centrifuge tube filled with a 75% ethanol solution [3]. Statolith pretreatment, embedding, grinding, polishing, and age counting were all carried out according to the methods described by Liu Bilin et al. [14]. The statolith increment pattern of the squid conformed to the “one-day” growth rule, and the hatching time was back-calculated based on the fishing time and age information from the statolith [3]. Stocks were divided into spring (hatched in the March–May period), summer (hatched in the June–August period), autumn (hatched in the September–November period), and winter (hatched in the December–February period) according to the hatching month [3]. In this study, according to the hatching period, the mantle length distribution, and the sex composition in each sampling year, 28, 26, and 28 statoliths were selected for the determination of trace elements from the summer, autumn, and winter stocks, respectively (Table 1).

#### 2.3.2. Determination of Statolith Trace Elements

The trace elements in the statoliths elements were analyzed at the Key Laboratory of the Ministry of Education for Sustainable Development of Fishery Resources, Shanghai Ocean University, via LA-ICP-MS. The laser ablation system was an UP-213, and an Agilent7700x device was used to carry out the ICP-MS [19]. During the laser ablation process, helium was used as the carrier gas and argon was used as the compensation gas to adjust the sensitivity [30]. The data for each sampling point included a blank signal of 20–30 S and a sampling signal of 50 S [30]. NIST610, MACS-3, BHVO-2G, and BIR-1G were used as calibration standard samples, and the element concentrations were analyzed quantitatively via external and internal standard methods [19]. ICPMS-Data-Cal software was used to process the sampling data offline to obtain the element concentrations [4].

Before the measurements were obtained, polished statolith slices were placed in an ultrasonic oscillator and mixed with ultrapure water for 5 min to remove impurities [22]. A sampling point was selected every 60 μm from the core to the edge of the statolith (the laser ablation diameter was 40 μm). An average of 10–11 sampling points were collected for each statolith (Figure 2). The growth (age) information contained in the laser ablation diameter corresponds to the water temperature of the habitat.

### 2.4. Statistical Analysis

The validity of the element determination results was evaluated according to the minimum detection limit for each element and the relative deviation (RSD%), which were calculated based on standard samples, and invalid elements were removed [31,32]. Data analyses were performed only for elements detected in all statoliths [31,31]. The axial distance from each measurement point to the core of the statolith was defined as the statolith diameter, and the ratios of each element to Ca at points which had the same statolith diameter in each sample were averaged to obtain the overall result [4,33]. The trend in the ratio of each element to Ca across stages of life history was evaluated based on means ± standard deviation. A one-way analysis of variance (ANOVA) was used to compare effective trace element concentrations among stocks [33]. Time series data for the effective trace elements and Ca ratios in the stocks were analyzed and compared via a one-way ANOVA [19]. A multivariate regression tree (MRT) model was used to cluster the ratios of key trace elements to Ca in the time series; in this model, the dependent variables were the values of the key trace elements/Ca ratios, and the independent variable was the diameter of the statolith [4]. According to the clustering results, the growth stages of the squid were determined, and changes in the ratios of the key trace elements to Ca across stages were analyzed. The multiple linear regression method was used to analyze the relationship between the fishing day temperature and the ratio of trace elements at the last point (20–30 d). The optimal model was selected according to the minimum Akaike information criterion (AIC), in which the dependent variable was the temperature variable, and the independent variable was the effective element ratio [5,19]. According to the relationship established above, the suitable temperature values and corresponding geographical locations for different life history stages were determined from the temperature database, and possible migration routes through all of the most suitable sea areas were determined for each growth stage [4,5,19].

### 2.5. Migratory Hypothesis and Remodeling

(1)The main assumptions of this research study are as follows [4,5,19].

The squid migrates in the South China Sea. According to the minimum AIC criterion, trace elements that are closely related to temperature variables were selected as indicator elements. It was assumed that the relationship between the trace elements and the temperature variables in different growth stages remained unchanged. To increase the range of predictability, all samples were used to establish the relationship between the temperature and the trace elements [4,5].

(2)The migration path remodeling process was as follows [4,5,19].

The hatching date was calculated by subtracting the age from the fishing date. The date for each growth stage was calculated as the hatching date plus the average age of the sampling points at each stage [3,4,5]. A regression analysis was used to establish the relationship between the ratios of trace elements and the temperature variables. To infer the migration route, the temporal resolution of the temperature data was set to one week, and the spatial resolution was set to 0.5° × 0.5°. The maximum swimming speed (20 km/d) of the squid was used to define the maximum range of movement [4,5]. Temperature variables and spatial ranges (longitude and latitude) corresponding to trace elements at different life history stages were determined. Sea areas with a range of suitable temperatures were identified as areas where migration may occur. Each sample had an optimal sea area. The sea area in which all samples appeared was the most likely sea area, and the probability of occurrence changed within the range of 0 to 1 [4,5,19].

(3)The calculation of the migration center of gravity [34].

Samples with a habitat probability greater than 0.6 at each stage of growth were selected to calculate the habitat center of gravity in order to analyze changes in the migration paths. The formula was as follows:$$G_{\mathrm{lon}}=\frac{\sum \left({\mathrm{Lon}}_{i}\times P_{i}\right)}{\sum P_{i}}$$
$$G_{\mathrm{lat}}=\frac{\sum \left({\mathrm{Lat}}_{i}\times P_{i}\right)}{\sum P_{i}}$$

In this formula, G_lon_ and G_lat_ are the longitude and latitude centroids of a certain growth stage, respectively; Lon_i_ and Lat_i_ are the longitude and latitude, respectively, at which the probability of a sample’s habitat at a certain stage is greater than 0.6; P_i_ is the probability of a certain sample’s habitat (greater than 0.6).

Figures were drawn using Microsoft Excel 2019. The ANOVA was implemented in SPSS 25. The multiple linear regression analysis was implemented in R using the “lm” package. The possible sea areas and their corresponding probabilities were calculated using the “geoR” package in R [4].

## 3. Results

### 3.1. Composition of Trace Elements

Five effective elements (except Ca), Na, Mg, Fe, Sr, and Ba, were identified in the statolith samples via LA-ICPMS. The minimum detection limits were 4.94, 7.81, 2.59, 167, 0, and 0 μmol/mol, and the measured values of the five elements were much higher than their lower detection limits (Table 2). The accuracy of the analysis based on standard samples was relatively high, with a relative standard deviation (% RSD) ranging from 1.44 (Na) to 5.23 (Mg).

The ANOVA showed that Mg, Fe, Sr, and Ba differed significantly among stocks (*P*_Mg_ < 0.05, F_Mg_ = 0.95, *n* = 738; *P*_Fe_ < 0.05, F_Fe_ = 10.13, *n* = 738; *P*_Sr_ < 0.05, F_Sr_ = 0.67, *n* = 738; *P*_Ba_ < 0.05, F_Ba_ = 1.16, *n* = 738). There was no significant difference in Ca and Na among the stocks (*p* > 0.05, *n* = 738). Therefore, the ratios of Mg, Sr, and Ba to Ca were used for subsequent analyses.

### 3.2. Trace Element Ratios

The Mg:Ca values in the summer, autumn, and winter stocks tended to decrease initially and increase thereafter (Figure 3), with peak values at the core (0 μm). At 0–180 μm, the Mg:Ca value of the summer stock was lower than the Mg:Ca values of the autumn and winter stocks, while at 360–480 μm, the Mg:Ca value of the summer stock was greater than the Mg/Ca values of the above two stocks. The Sr:Ca value decreased initially and then increased in the summer and winter stocks (Figure 3), while it showed a downward trend in the autumn stock (Figure 3). All three stocks showed a peak at the core (0 μm). At 0–180 μm, the Sr:Ca value of the autumn stock was greater than the Sr:Ca values of the summer and winter stock, while at 300–480 μm, the value for the autumn stock was smaller than those of the above two stocks. At 120–300 μm, the Sr:Ca value of the summer stock was greater than that of the winter stock, while at 360–540 μm, the Sr:Ca value of the summer stock was smaller than that of the winter stock. The Ba:Ca values in the summer, autumn, and winter stocks showed an overall upward trend (Figure 3) and reached peaks at 480 μm, 420 μm, and 540 μm, respectively. At 240–420 μm, the Ba:Ca value of the autumn stock was greater than the Ba:Ca values of the summer and winter stocks.

The ANOVA showed that there were significant differences in element ratios between the winter stock and the summer and autumn stocks (*P_summer_*
_Mg:Ca_ < 0.05, F_summer Mg:Ca_ = 11.41, *n* = 504; *P_summer_*
_Sr:Ca_ < 0.05, F_summer Sr:Ca_ = 25.45, *n* = 504; *P_summer_*
_Ba:Ca_ < 0.05, F_summer Ba:Ca_ = 12.34, *n* = 504; *P_autumn_*
_Mg:Ca_ < 0.05, F_autumn Mg:Ca_ = 13.14, *n* = 486; *P_autumn_*
_Sr:Ca_ < 0.05, F_autumn Sr:Ca_ = 10.61, *n* = 486; *P_summer_*
_Ba:Ca_ < 0.05, F_autumn Ba:Ca_ = 11.67, *n* = 486). There were no significant differences in the Sr:Ca and Ba:Ca ratios between the summer and autumn stocks (*p* > 0.05, *n* = 486).

### 3.3. Multivariate Time Series Clustering

The MRT showed (Figure 4) that the otoliths of the three stocks could be divided into four nodes from the core to the edge in the statolith and formed five clusters. The statolith nodes of the summer stock were 109 μm, 262 μm, 335 μm, and 437 μm; the statolith nodes of the autumn stock were 89 μm, 231 μm, 285 μm, and 343 μm; and the statolith nodes of winter stock were 107 μm, 163 μm, 255 μm, and 403 μm. According to the clustering results for the trace elements in the statoliths, the squid could be divided into an embryonic stage (cluster 1), larval stage (cluster 2), juvenile stage (cluster 3), subadult stage (cluster 4), and adult stage (cluster 5), and the clustering patterns of the summer and autumn stocks were similar.

According to the clustering results, the summer, autumn, and winter stocks were divided into five life history stages, and the ratios of the key trace elements to Ca at different growth stages differed among the stocks (Figure 5, Table 3). In the summer stock (Figure 5a), Sr:Ca decreased from the embryonic stage to the adult stage, and Ba:Ca decreased from the embryonic stage to the juvenile stage and increased from the juvenile stage to the adult stage. In the autumn stock (Figure 5b), both the Mg:Ca and Sr:Ca ratios decreased from the embryonic stage to the adult stage, and the Ba:Ca ratio decreased from the embryonic stage to the larval stage. In the winter stock (Figure 5c), the Mg:Ca ratio showed a decreasing trend from the embryonic stage to the adult stage, and the Sr:Ca and Ba:Ca ratios showed decreasing trends from the embryonic stage to the sub-adult stage.

The ANOVA showed that there were significant differences in the Mg:Ca, Sr:Ca, and Ba:Ca ratios among growth stages in the summer stock (*P*_Mg:Ca_ < 0.05, F_Mg:Ca_ = 3.82, *n* = 140; *P*_Sr:Ca_ < 0.05, F_Sr:Ca_ = 3.07, *n* = 140; *P*_Ba:Ca_ < 0.05, F_Ba:Ca_ = 3.59, *n* = 140), autumn stock (*P*_Mg:Ca_ < 0.05, F_Mg:Ca_ = 5.46, *n* = 130; *P*_Sr:Ca_ < 0.05, F_Sr:Ca_ = 18.92, *n* = 130; *P*_Ba:Ca_ < 0.05, F_Ba:Ca_ = 11.01, *n* = 130), and winter stock (*P*_Mg:Ca_ < 0.05, F_Mg:Ca_ = 7.19, *n* = 140; *P*_Sr:Ca_ < 0.05, F_Sr:Ca_ = 2.91, *n* = 140; *P*_Ba:Ca_ < 0.05, F_Ba:Ca_ = 4.83, *n* = 140).

### 3.4. Relationship between Element Ratios and Temperature Variables

A multiple linear regression analysis was performed to evaluate the relationships between the ratios of environmental variables and the trace elements in each stock. As shown in Table 4, in the winter stock, Temp_25 and the Sr:Ca ratio had the best fit, with a model R^2^ and AIC values of 0.33 and 11.85, respectively, and the Sr/Ca ratio had a significant impact (*p* < 0.05). In the summer stock, T the best fit model for Temp_25 included the Mg:Ca, Sr:Ca, and Ba:Ca ratios, with model R^2^ and AIC values of 0.55 and 25.03, respectively, and the Mg:Ca, Sr:Ca, and Ba:Ca ratio variables had a significant impact (*p* < 0.01). In the autumn stock, the best-fitting model for Temp_25 included the Sr:Ca and Ba:Ca ratios, with model R^2^ and AIC values of 0.40 and 5.13, respectively, and the Sr:Ca and Ba:Ca ratios were significant factors (*p* < 0.05). The formulae for the best fit models were as follows:Winter stock: Temp_25 = 33.12 − 0.03 Sr:Ca
Summer stock: Temp_25 = 27.19 + 0.0006 Mg:Ca + 0.0031 Sr:Ca − 0.2019 Ba:Ca
Autumn stock: Temp_25 = 23.80 + 0.02 Sr:Ca + 2.79 Ba:Ca

### 3.5. Distribution of Migratory Habitats

According to the best-fitting model for the environmental variables and trace element ratios, the temperature for different life stages was inversely deduced (Table 5). The clustering of life stages and the trend in the Sr:Ca ratio for the summer and autumn stocks were similar. Therefore, these groups were integrated to reconstruct the migration path. The water temperature of the winter stock (25 m) was 28.05–28.88 °C at the embryonic stage, 28.16–28.94 °C at the larval and juvenile stages, and 28.05–28.97 °C at the adult stage. The water temperature (25 m) of the summer–autumn stock was 27.38–27.92 °C at the embryonic stage, 27.01–27.81 °C at the larval and juvenile stages, and 26.92–27.73 °C at the adult stage.

According to the water temperature of the stock habitat, the potential area for migration in the South China Sea was determined (Figure 6, Figure 7 and Figure 8). The centers of distributions for the winter stock were 114.52° E, 9.17° N at the embryonic stage; 115.71° E, 7.96° N at the larval stage; 112.42° E, 10.63° N at the juvenile stage; 112.09° E, 14.99° N at the sub-adult stage; and 110.10° E, 15.68° N at the adult stage. After the winter stock hatched in the southern South China Sea, the stock moved to the Northwest Sea for nursing and feeding. The centers of distributions for the summer–autumn stock were 113.50° E; 17.70° N at the embryonic stage; 113.29° E, 17.14° N at the larval stage; 113.58° E, 14.83° N at the juvenile stage; 114.99° E, 14.17° N at the sub-adult stage; and 116.02° E, 12.47° N at the adult stage. After the summer–autumn stock hatched in the northern part of the South China Sea, there was a shift to the southeast for nursing and feeding. Therefore, there were differences in habitat among the stocks at different life stages.

## 4. Discussion

### 4.1. Analysis of Trace Elements

This study showed that the statolith concentrations of Mg, Sr, and Ba differ significantly among stocks, and these differences may be related to differences in seawater temperature, food source, and salinity, which affect the deposition of trace elements [22,28]. Sr and Ba are important nutrients for individual metabolism [33,35]. The Sr content reflects the temperature and salinity of the habitat, and Ba is an effective indicator of vertical swimming by squid and ocean upwelling [15,36]. Na and Mg are important elements in the biomineralization process of stones, where the concentration of Mg is related to the precipitation of organic matter and is highly physiologically regulated [12,32].

In addition, the diurnal vertical movements of individuals and changes and transitions in feeding behaviors also affect the composition of trace elements in the population [3,26,27,29]. Under recent global climate change, annual seasonal changes in marine environmental conditions may also affect the absorption of trace elements by populations, and the seawater temperature affects the pH value of the blood and endolymphatic fluid system [37,38,39]. These changes affect the transport of trace elements, thereby including the screening of trace elements by lymph [37]. Han et al. showed that the ratios of Sr:Ca and Ba:Ca are important indicators for the construction of habitats and migration of *Ommastrephes bartramii* by establishing a relationship between the ratio of trace elements in the statolith element ratio and environmental factors, and the Ba:Ca ratio characterizes the vertical migration ability of squid [4]. Therefore, the concentrations of different types of trace elements are regulated by physiological processes and can reflect changes and variations in the marine environment, including seasonal changes in marine elements [22,28].

### 4.2. Analyses of Effective Trace Elements Relative to Ca

The ratios of trace elements to Ca in statolith samples are an important indicator of habitat changes at different life stages and provide an effective basis to infer life history traits and reconstruct migration routes [6,19,33]. Generally, the Sr:Ca and Ba:Ca ratios in cephalopod statoliths have a negative correlation with water temperature, Sr/Ca is correlated with salinity, and Ba/Ca has a positive correlation with water depth (providing a useful indicator of the vertical movement of cephalopods) [5,16,40]. In this study, the Mg:Ca ratios of the summer, autumn spawning, and winter stocks decreased throughout the life cycle within 0–480 μm. The concentration of Mg is related to the deposition of organic matter in the statolith. As an individual grows and develops, the proportion of organic matter decreases gradually, which may lead to a decrease in the Mg:Ca ratio in the statolith [11,41,42]. In addition, Zumholz et al. found that the Mg/Ca value of squid tends to decrease from the core to the edge of the statolith, which is believed to be related to the growth rate of the statolith and the grade of gonad maturity [15]. The Ba:Ca values in the summer, autumn, and winter stocks decreased initially (0–360 μm) and then increased (360–600 μm). Studies have shown that the egg capsules of Ommastrephidae are gelatinous masses (containing thousands of eggs) which form large floating bodies, usually staying at a certain depth (related to density), and hatched larvae gradually rise to the surface. Adults will move to deeper waters [12,43,44]. In addition, the water layers inhabited by different stocks are related to the abundance of prey, and the range of diurnal vertical movement and feeding conditions among groups are also related to the distribution of water layers of stocks, which can affect Ba:Ca values [4,22,27].

The Sr:Ca value decreased and then increased in the summer and winter stocks but showed a downward trend in the autumn stock, which may be related to the biotic and abiotic effects of the habitat at stages of life history [28]. The summer and autumn stocks may experience a higher water temperature in the early stage, which may lead to an early decrease in the Sr:Ca ratio; as the feeding ability improves, the Sr:Ca value is affected by biotic and abiotic factors [3,22,28]. The roles of abiotic factors and the relative contributions of different factors may lead to changes in the ratios of elements in the summer and autumn stocks in the late life cycle [3]. The Sr:Ca ratio of the winter stock decreased initially and then increased, and this was related to the slow warming experienced by the population in the early stage [28]. As the gonads gradually mature, the influence of elements absorbed by the stock in the sub-adult stage may be greater than that of the water temperature, explaining the increase from the sub-adult stage to the adult stage [22,28]. Therefore, the Mg:Ca, Sr:Ca, and Ba:Ca ratios can be used as indicators to characterize habitat changes during the life cycle of *S. oualaniensis* in the South China Sea. The differences in element ratios among stocks may be related to differences in habitat, and the relative contributions of biotic and abiotic factors to variations in element ratios may also differ among life history stages [3,22,26,29].

### 4.3. Habitat Changes across Life History Stages

In the larval stage, cephalopods passively migrate with ocean currents [5,19]. As gonads mature, the active swimming ability increases and the population migrates over long distances for feeding to meet their physiological needs [3,4]. It is the largest semi-enclosed deep-water basin in the western Pacific, a typical monsoon region, and has significant local air–sea interactions [29]. The surface and upper-level circulation of the South China Sea is significantly affected by monsoons [45,46]. Driven by the southwest monsoon in the summer, the surface circulation of the South China Sea presents a strong anticyclonic structure in the south and weak cyclonic circulation in the northern part of the South China Sea [46,47,48]. Driven by the northeast monsoon in the winter, the surface circulation structure in the South China Sea is cyclonic and obviously strengthens the west boundary current flowing south along the coast of Vietnam [48]. Spring and autumn are transition periods for the South China Sea monsoons [47,48]. The corresponding circulation characteristics also exhibit a transitional flow pattern between winter and summer circulation, and the flow velocity is weaker than that in the winter and summer [46,48]. In addition, the exchange of water between the South China Sea and the outside ocean through various strait passages impacts circulation [45]. The Kuroshio Current has an important impact on the circulation pattern in the northern part of the South China Sea, causing the Guangdong Coastal Current, Kuroshio Intrusion Current, Dongsha Current, South China Sea Warm Current, and Luzon Current to jointly form the upper-level current field in the northern part of the South China Sea [45,47]. Therefore, the configuration and properties of circulation in the South China Sea and its monsoon climate are related to variations in the habitats and life histories of marine taxa [48].

Generally, early larvae exhibit active feeding, with a focus on crustaceans, and their feeding habits change from zooplankton to nocturnal free-swimming fish [20,22,29]. According to the MRT in this study, the ratios of the effective elements in each stage of growth differed among stocks. There were habitat differences between the winter and summer–autumn stocks, further increasing the difference in the spatial distribution of the stocks. Based on an acoustic analysis, Zhang et al. found that in the South China Sea, under natural conditions, the 10–50 m layer is the water layer in which squid mainly gather, and this matches the 25 m water layer chosen in this study [49]. These findings indicate that this water layer is a suitable habitat for the squid population.

The Guangdong Coastal Current, Kuroshio Intrusion Current, Dongsha Durrent, and South China Sea Warm Current are important current systems in the South China Sea [46]. Seasonal changes in ocean current attributes directly affect the hydrological conditions, water mass configuration, circulation structure, and fishing ground locations in the South China Sea [45,46]. In this study, the hatching water temperature of the winter stock was 28.05–28.88 °C (Table 5), and, according to the model, the stock hatched in the southern part of the South China Sea (Figure 6 and Figure 8). The hatched individuals may move southward with the cyclonic circulation under the influence of the northeast monsoon in the winter. The larvae then move northwest under the action of the summer monsoon and grow and develop in the anticyclonic circulation [22,28,45,46]. In addition, the hatching water temperature of the summer–autumn stock was 27.38–27.92 °C (Table 5), and the model predicted that this population hatches in the northern part of the South China Sea (Figure 7 and Figure 8). The hatched individuals may move in the mesoscale closed anticyclonic circulation in the northern South China Sea [22,28]. The stock is affected by the winter monsoon, and larvae move to the southern part of the South China Sea [28,45,48]. During individual life cycles, winter and summer–autumn stocks are fished in the autumn and winter, respectively; the temperature at each growth stage is inversely related to the water temperature on the fishing day, and the temperature difference in fishing seasons may cause the water temperature of the habitat of the winter stock to be higher than that of the habitat of the summer–autumn stock [3]. Therefore, differences in the migration routes of the winter and summer–autumn stocks are significantly affected by the monsoon climate and cyclone circulation in the sea area [46,48].

## 5. Conclusions

Na, Mg, Fe, Sr, and Ba are effective elements for analyzing population histories and migration in the statoliths of *S. oualaniensis*. Throughout the stages of life history, the Mg:Ca ratio in the population showed a decreasing trend, while the Sr:Ca and Ba:Ca ratios showed initial decreases, followed by increasing trends. Based on the MRT results and hatching temperature, the summer and autumn stocks may share similar habitats. A linear relationship between Temp_25 and the ratios of Mg:Ca, Sr:Ca, and Ba:Ca was the best model for inferring the migration route of the summer stock, while the linear relationship between Temp_25 and Sr:Ca was the best model for the winter stock. The hatching water temperatures of winter and summer–autumn spawning stocks were 28.05–28.881 °C and 27.15–27.92 °C, respectively. Based on the water temperature at each growth stage and the model results, the winter stock hatched in the southern South China Sea, after which the larvae moved to the northwestern part of the sea area under the action of the summer monsoon and grew and developed in the anticyclonic circulation. The summer–autumn spawning stock hatched in the northern South China Sea, and the larvae migrated in the mesoscale closed anticyclonic circulation in the northern South China Sea. The migration route is significantly affected by the seasonal circulation in the South China Sea, and monsoon and eddy characteristics are important factors affecting the direction of the migration of individuals.

## Figures and Tables

**Figure 1 animals-13-02811-f001:**
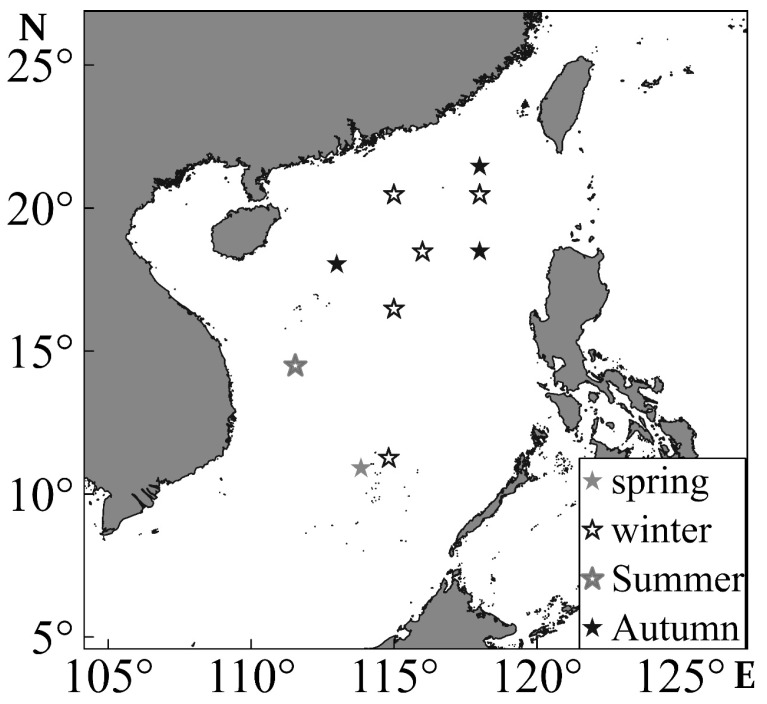
*Sthenoteuthis oualaniensis* sampling site in the South China Sea. Squares, circles, triangles, and diamonds represent the spring, summer, autumn, and winter sampling sites, respectively.

**Figure 2 animals-13-02811-f002:**
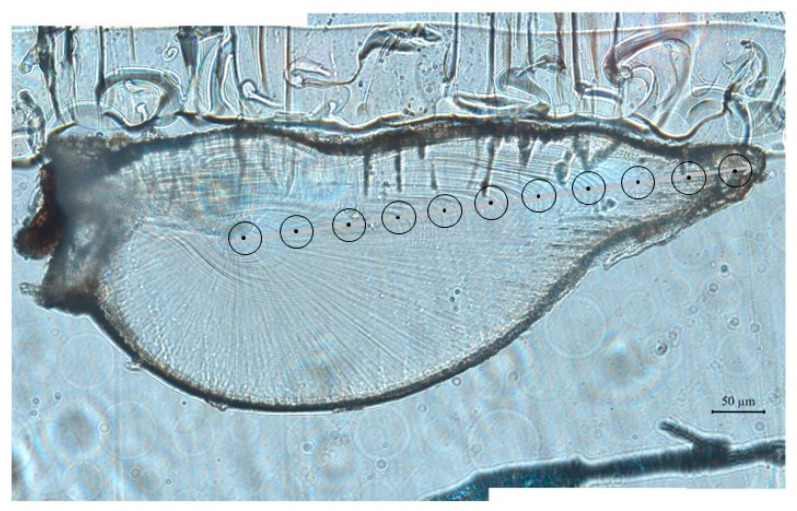
Analytic spots (·) in a statolith from *S. oualaniensis*.

**Figure 3 animals-13-02811-f003:**
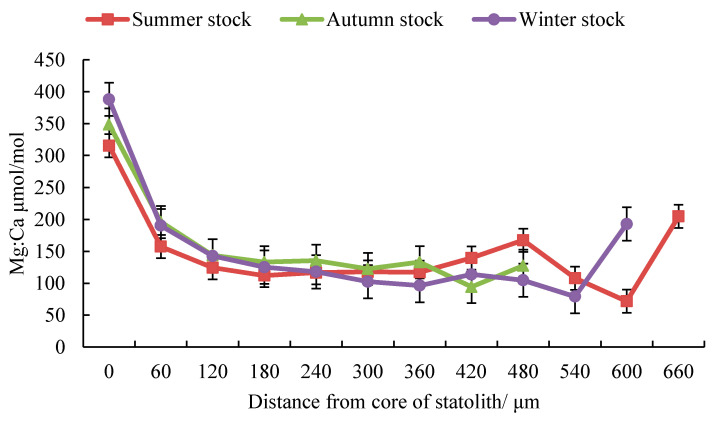
Change in effective elements ratio in statoliths from *S. oualaniensis* in a time series.

**Figure 4 animals-13-02811-f004:**
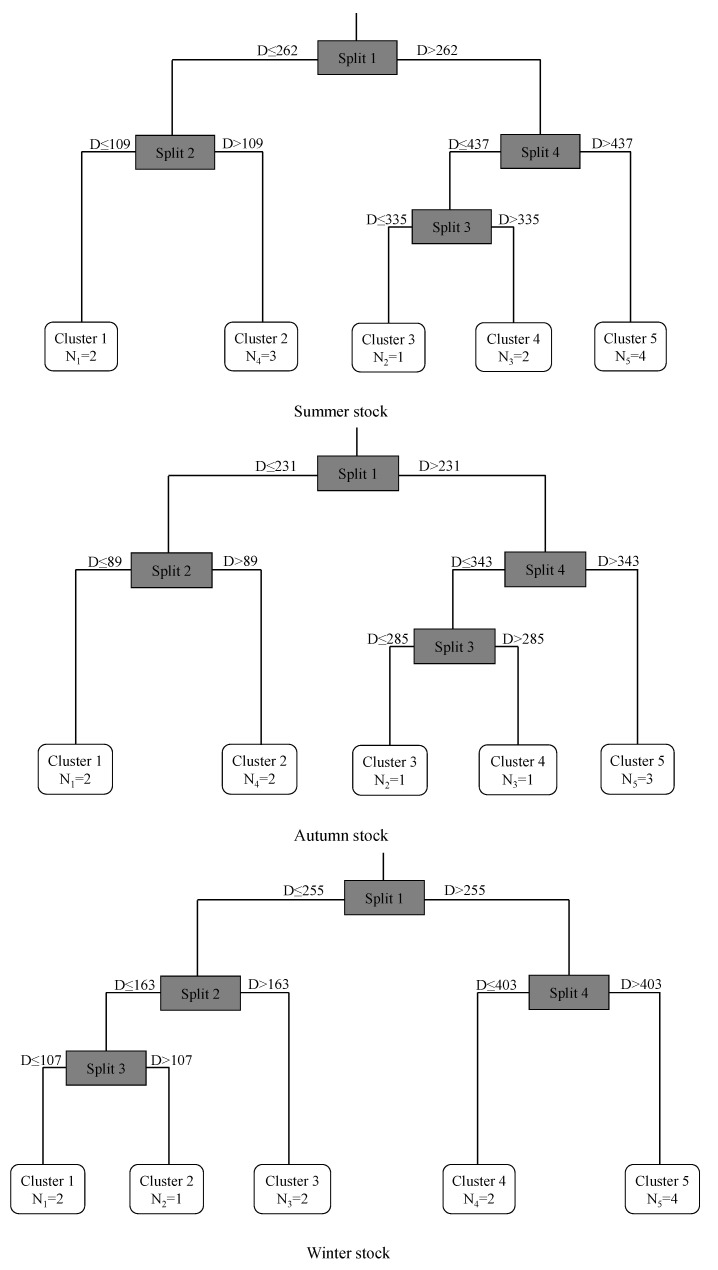
Trace element clustering of statoliths from *S. oualaniensis* based on a multiple regression tree model.

**Figure 5 animals-13-02811-f005:**
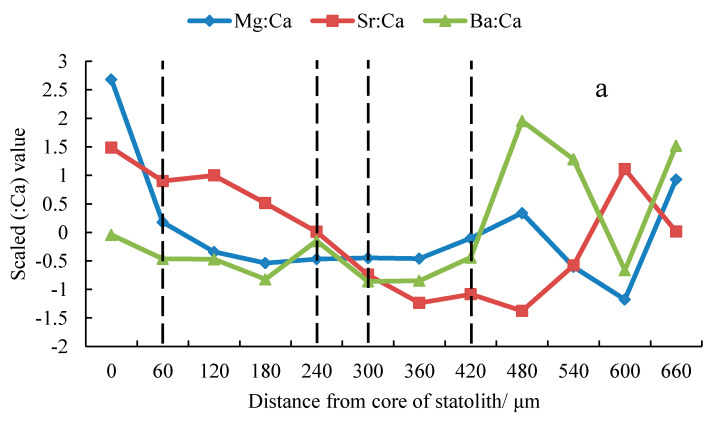
The ratios of key trace elements to calcium in the statoliths in different clusters. (**a**–**c**) Show the summer, autumn, and winter stocks, respectively.

**Figure 6 animals-13-02811-f006:**
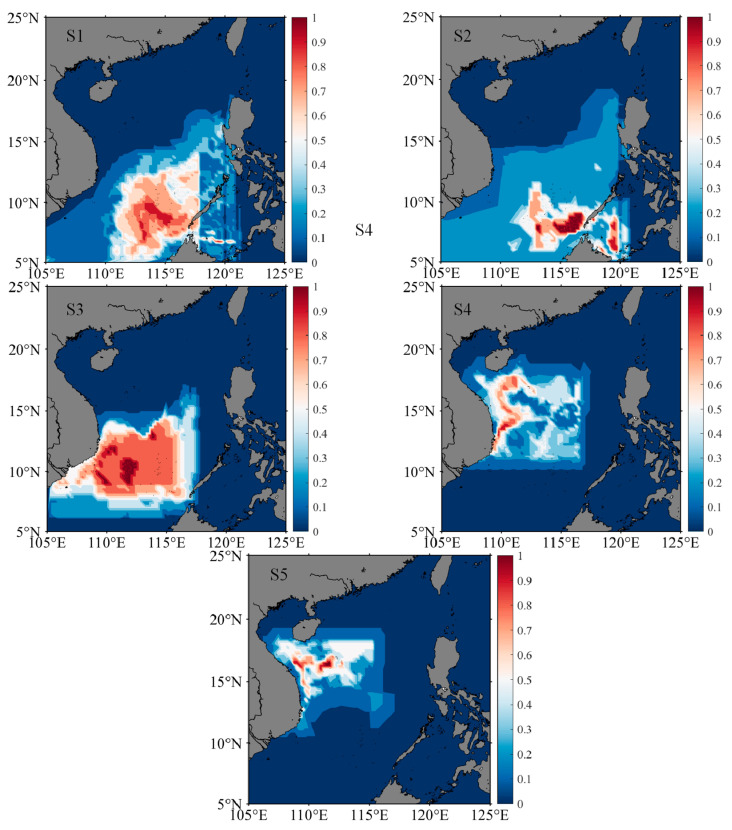
Potential distribution areas of winter stock at different growth stages. S1, S2, S3, S4, and S5 represent individual embryonic, larval, juvenile, sub-adult, and adult stages, respectively. Heat bars represent the probability that the stock is in the predicted sea area.

**Figure 7 animals-13-02811-f007:**
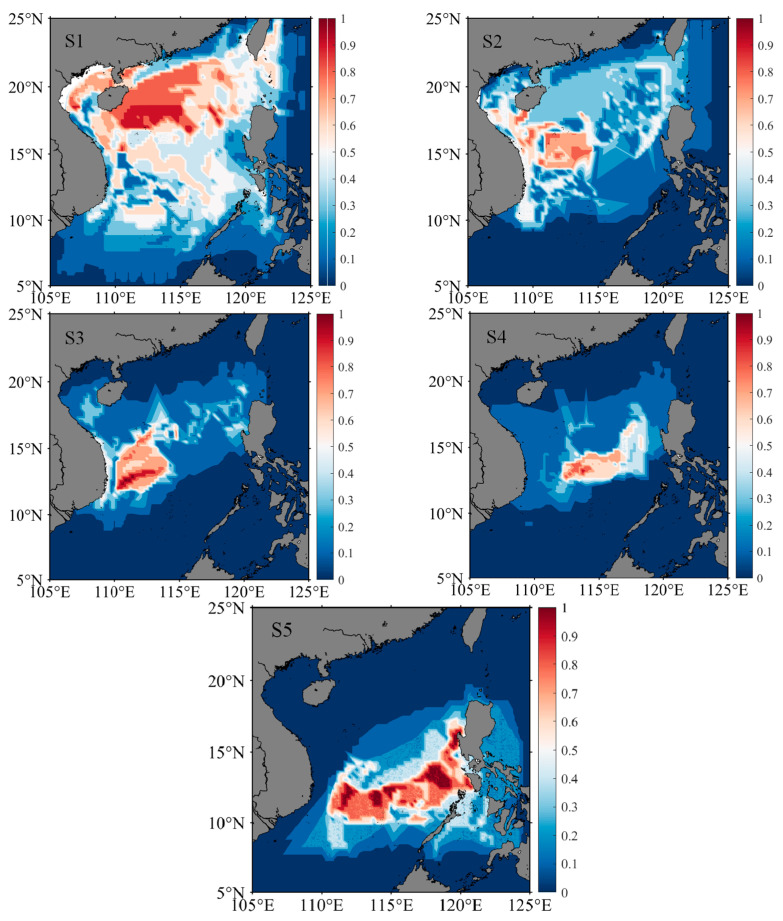
Potential distribution areas of summer–autumn stock at different growth stages. S1, S2, S3, S4, and S5 represent individual embryonic, larval, juvenile, sub-adult, and adult stages, respectively. Heat bars represent the probability that the stock is in the predicted sea area.

**Figure 8 animals-13-02811-f008:**
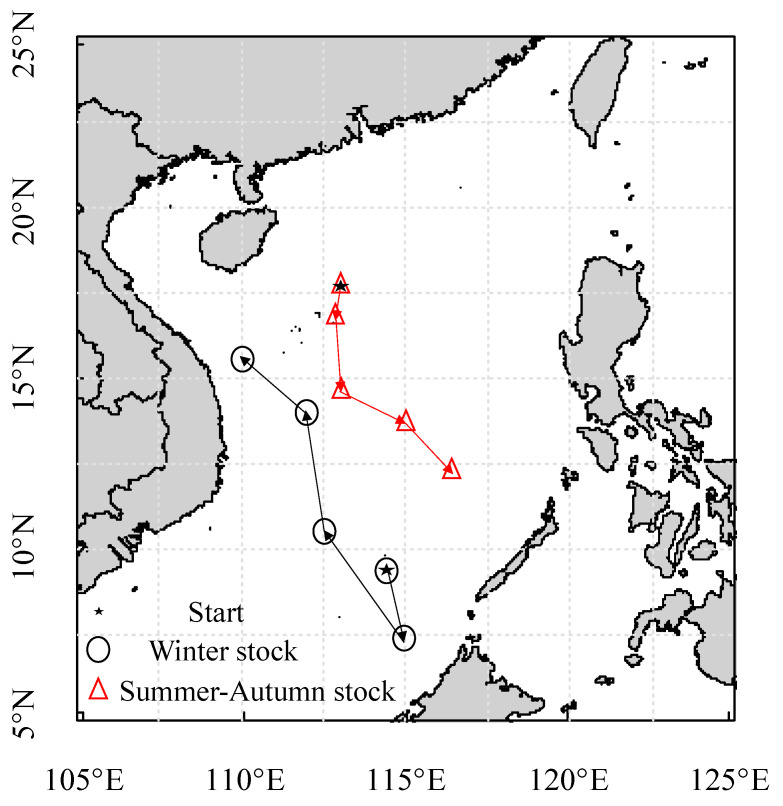
Spatial distribution of migration centers at different growth stages of winter and summer–autumn stocks.

**Table 1 animals-13-02811-t001:** Information from *S. oualaniensis* statolith samples tested via microchemical analysis.

Stocks	Sampling Seasons	Measurement Quantities	Mantle Length/mm	Age/Day	Weight/g	Gonad Maturity	Sex Ratios F/M
Winter	Autumn	28	119–142	184–223	29–195	I–IV	0.6:1
Summer	Spring	28	103–152	172–259	26–1109	I–IV	1:1
Autumn	Summer	26	98–124	182–244	46–131	I–IV	1:1

**Table 2 animals-13-02811-t002:** Effective trace element concentrations in statoliths from *S. oualaniensis.*

Stock		Concentrations/μmol/mol					
		Na	Mg	Ca	Fe	Sr	Ba
Summer stock	Min	3270.39	2238.22	40,933,039.82	63.86	4925.29	2.93
	Max	5479.59	50,109.44	41,209,251.08	1185.97	7711.54	53.64
	Mean	4266.74	6818.52	41,103,854.93	348.02	6239.47	6.32
	Std	532.22	6580.35	58,092.83	360.85	441.79	5.97
Autumn stock	Min	3670.79	3133.53	40,942,215.57	61.93	5415.46	3.78
	Max	5122.58	118,998.03	41,198,206.00	432.18	7842.94	17.32
	Mean	4385.96	8809.93	41,099,665.07	169.84	6313.49	6.09
	Std	321.13	15,931.08	50,312.46	103.90	506.55	2.04
Winter stock	Min	3315.17	2731.72	40,964,425.73	64.70	5519.29	2.75
	Max	6067.07	31,658.01	41,216,403.36	553.18	7062.98	10.28
	Mean	4238.17	6535.40	41,120,434.76	210.02	6224.37	5.38
	Std	501.68	5517.04	52,934.93	127.05	387.61	1.48

**Table 3 animals-13-02811-t003:** Ratios of key trace elements to calcium at different life history stages.

Stock	Stage	Distance	Number Spots	Mg:Ca μmol/mol		Sr:Ca μmol/mol		Ba:Ca μmol/mol	
				Mean	Std	Mean	Std	Mean	Std
Summer	Embryonic	D ≤ 109	2	236.42	122.58	158.08	6.49	0.13	0.02
	Larval	109 < D ≤ 262	3	117.70	23.71	154.54	11.52	0.12	0.02
	Juvenile	262 < D ≤ 335	1	117.83	41.21	148.08	6.91	0.11	0.02
	Sub-adult	335 < D ≤ 437	2	128.38	47.05	145.90	3.88	0.12	0.02
	Adult	D ≥ 437	4	139.23	45.02	145.38	9.71	0.21	0.11
Autumn	Embryonic	D ≤ 89	2	243.88	97.77	170.08	7.76	0.15	0.01
	Larval	89 < D ≤ 231	2	138.36	18.89	156.78	5.24	0.12	0.01
	Juvenile	231 < D ≤ 285	1	135.33	28.78	149.49	4.58	0.13	0.01
	Sub-adult	285 < D ≤ 343	1	122.38	19.71	148.34	3.73	0.13	0.04
	Adult	D ≥ 343	3	121.08	33.91	143.13	4.58	0.14	0.02
Winter	Embryonic	D ≤ 107	2	289.29	168.24	158.68	8.87	0.14	0.02
	Larval	107 < D ≤ 163	1	142.77	25.20	154.61	8.31	0.14	0.01
	Juvenile	163 < D ≤ 255	2	121.60	23.59	147.88	5.41	0.12	0.01
	Sub-adult	255 < D ≤ 403	2	99.44	21.08	146.30	6.57	0.10	0.01
	Adult	D ≥ 403	4	102.67	22.47	153.51	9.44	0.14	0.03

**Table 4 animals-13-02811-t004:** Fitting relationship between the ratios of trace elements to Ca and the temperature variables in different stocks. The temperature was evaluated at different water layers, including depths of 5 meters (Temp_5), 25 meters (Temp_25), 55 meters (Temp_55), 75 meters (Temp_75), and 105 meters (Temp_105).

Stocks	Environment Variables	Elements Ratios	Equations	R^2^	AIC	P	Sign
Winter	Temp_5	Sr:Ca	T5 = 31.56 − 0.02 Sr:Ca	0.22	17.85	*p* > 0.05	ns
	Temp_25	Sr:Ca	T25 = 33.12 − 0.03 Sr:Ca	0.33	11.85	* p * < 0.05	*
	Temp_55	Sr:Ca	T55 = 46.12 − 0.14 Sr:Ca	0.10	43.28	*p* > 0.05	ns
	Temp_75	Sr:Ca	T75 = 45.23 − 0.15 Sr:Ca	0.10	44.07	*p* > 0.05	ns
	Temp_105	Sr:Ca	T105 = 31.51 − 0.08 Sr:Ca	0.13	32.47	*p* > 0.05	ns
Summer	Temp_5	Ba:Ca	T5 = 27.72 + 0.18 Ba:Ca	0.34	27.56	*p* < 0.05	*
	Temp_25	Mg:Ca Sr:Ca Ba:Ca	T25 = 27.19 + 0.0006 Mg:Ca + 0.0031 Sr:Ca − 0.2019 Ba:Ca	0.75	25.03	* p * < 0.01	**
	Temp_55	Sr:Ca	T55 = 27.27 − 0.0003 Sr:Ca	0.12	52.95	*p* > 0.05	ns
	Temp_75	Ba:Ca	T75 = 26.04 + 1.12 Ba:Ca	0.03	32.29	*p* > 0.05	ns
	Temp_105	Ba:Ca	T105 = 21.54 + 1.13 Ba:Ca	0.10	38.53	*p* > 0.05	ns
Autumn	Temp_5	Mg:Ca	T5 = 28.29 + 0.0054 Mg:Ca	0.12	14.75	*p* > 0.05	ns
	Temp_25	Sr:Ca Ba:Ca	T25 = 23.80 + 0.02 Sr:Ca + 2.79 Ba:Ca	0.40	5.13	* p * < 0.05	*
	Temp_55	Mg:Ca	T55 = 25.35 − 0.02 Mg:Ca	0.16	29.31	*p* > 0.05	ns
	Temp_75	Mg:Ca	T75 = 23.64 − 0.03 Mg:Ca	0.17	33.38	*p* > 0.05	ns
	Temp_105	Mg:Ca	T105 = 19.79 − 0.02 Mg:Ca	0.18	29.33	*p* > 0.05	ns

Note: Underline means the suitable environmental variables with the lowest AIC; * means significant difference; ** means very significant difference.

**Table 5 animals-13-02811-t005:** Potential habitat water temperature of different clusters (stocks).

Growth Stage	Winter Stock	Summer Stock	Autumn Stock
	Temperature/°C	Temperature/°C	Temperature/°C
Embryonic	28.05–28.88	27.70–27.92	27.38–27.92
Larval	28.16–28.89	27.65–27.81	27.15–27.51
Juvenile	28.45–28.94	27.64–27.76	27.01–27.27
Sub–adult	28.27–28.97	27.67–27.73	26.98–27.45
Adult	28.05–28.91	27.65–27.72	26.92–27.20

## Data Availability

The data that support the findings of this study are available from the corresponding author upon reasonable request.

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
