# Peer review of "Migration Route of Sthenoteuthis oualaniensis in the South China Sea Based on Statolith Trace Element Information"

_animals, 2023, doi:10.3390/ani13182811_

Round 1

Reviewer 1 Report

Overall a good paper with some points that could be considered for further improvement:

Line 95: Why are the months not arranged chronologically if all sampling was done in the same year (2020)?

Lines 106-108: This section should be moved from "Materials and Methods" to "Discussion".

Line 131: In Table 1, indicate what the "Quantities" column refers to.

Species names should be italicised (in lines: 23, 103, 149, 241).

In the text, the citation should be numbered (with reference numbers in square brackets), according to the Journal's Guide for Authors.

Reviewer 2 Report

The composition ratio of key elements of the Iris squid's otoliths is used in this paper to explore the Iris squid's migration route, to understand its life history characteristics, and to fit the model with temperature to speculate its migration route, and the study's conclusions provide scientific guidance for the sustainable development and utilization of Iris squid resources. The study effort is novel, the data analysis is more comprehensive, the data interpretation is more appropriate, and the writing is more consistent. However, there are still a few small issues.

1Individual legends in Figure 1 cannot be clearly seen, and it is suggested to adjust the clarity map appropriately.

2The inconsistency of sampling points, whether it affects the analysis of the results, only the winter sampling point covers a line, spring and autumn are only 1 point, and the spring data are not reflected in the full text. It is recommended that the spring data be supplemented or the sampling map be modified.

3In the Migratory Hypothesis and Remodeling, in the section on model construction, "To infer the migration route, the temporal resolution of the temperature data was set to infer the migration route, the temporal resolution of the temperature data was set to one week, and the spatial 196 resolution was set to 0.5° × 0.5°." Does it make sense to pick a one-week period for the temperature data? Does the breadth cover the corresponding variation

4 In Distribution of Migratory Habitats, the results above are arranged according to summer, autumn, and winter, but Figures 6-8 are not arranged in this order, so it is suggested to unify the description of the results in the whole text.

5Unified format for charts, annotations, and titles in the text. As shown in Figure 6 and Table 4.

No problem

Reviewer 3 Report

The introduction and discussion could be better including classical literature about cephalopod migration patterns.

Literature recommended:

Boyle P, & Rodhouse P (2005) Cephalopods, ecology and fisheries. Blackwell, London

Gilly, W.F., U. Markaida, C.H. Baxter, B.A. Block, A. Boustany, L. Zeidberg, K. Reisenbichler, B. Robison, G. Bazzino & C. Salinas. 2006. Vertical and horizontal migrations by the squid Dosidicus gigas revealed by electronic tagging. Mar. Ecol. Prog. Ser., 324: 1-17.

Hatanaka, H., S. Kawahara, Y. Uozumi & S. Kasahara, 1985. Comparison of life cycles of five ommastrephid squid fished by Japan: Todarodes pacificus, Illex illecebrosus, Illex argentinus, Nototodarus sloani sloani and Nototodarus sloani gouldi. NAFO Scientific Council Studies, 9: 59-68.

Hurley, G.V & E.G. Dawe, 1981. Tagging studies on squid (Illex illecebrosus) in the Newfoundland area. NAFO Scr. Doc., 80/II/33.

Sakai, M., Tsuchiya, K., Mariategui, L., Wakabayashi, T., & Yamashiro, C. (2017). Vertical Migratory Behavior of Jumbo Flying Squid (Dosidicus gigas) off Peru: Records of Acoustic and Pop-up Tags. Japan Agricultural Research Quarterly: JARQ, 51(2), 171-179.

Semmens, J. M., Pecl, G. T., Gillanders, B. M., Waluda, C. M., Shea, E. K., Jouffre, D., ... & Shaw, P. W. (2007). Approaches to resolving cephalopod movement and migration patterns. Reviews in Fish Biology and Fisheries, 17(2-3), 401.

Stewart, J. S., Hazen, E. L., Foley, D. G., Bograd, S. J., & Gilly, W. F. (2012). Marine predator migration during range expansion: Humboldt squid Dosidicus gigas in the northern California Current System. Marine Ecology Progress Series, 471, 135-150.

Yatsu, A., K. Yamanaka & C. Yamashiro. 1999. Tracking experiments of the Jumbo Flying Squid, Dosidicus gigas, with an ultrasonic telemetry system in the Eastern Pacific Ocean. Bulletin of the National Research Institute of far Seas Fisheries, 36: 55-60.

The sample size is small, and authors must be more cautious with statistical analyses and their interpretation. Power test could be useful to estimate the power of the analyses used (i.e., anova).

ANOVA results will better explain in a Table. Please include a table with a sample size.

Please start the discussion section with a short sentence about the principal results of the research and later compare it with other papers.

Take care with the concepts of life history and life cycle because there are no synonyms.

See Ibáñez CM (2020) Sobre el uso de los conceptos de ciclo de vida e historia de vida en ecología y evolución. Gayana 84(2): 93-100. http://dx.doi.org/10.4067/S0717-65382020000200093

https://www.scielo.cl/scielo.php?script=sci_arttext&pid=S0717-65382020000200093

some sentences are disconnected, please improve it.

Reviewer 4 Report

Your paper suffers from the poor quality of the presentation of science being the topic of your MS. Syntax of many of your sentences is twisted and convoluted, leaving an impression of simple matters being unduly complicated. First remedy: give your text to the native English speaker, preferably familiar with the topic of your study. Second remedy: eliminate these sentences which are not directly connected with your study (most of them are broad & trivial). Third remedy: tighten your scientific argument & logic. Fourth remedy: Your referencing is inadequate. Please revise. Your paper is moderately successful attempt to link environment and life cycle in a specific model. Yet, you do not attempt to revise staging of the ommastrephid squid life cycle(s). It is worth doing!

Please look at the attached file.

Syntax! Low clarity.

Reviewer 5 Report

The aim of this study is to uncover the migration route of Sthenoteuthis oualaniensis in the South China Sea with tracing the records of some elements in the statolith. It shows useful and important information about the dynamics of fisheries resource that can be affected by the oceanic environmental factors. The strength of this study is to detect the environmental history that the squid experienced as the material evidences.

As summarized below, there are several concerns in this manuscript. They are mainly about the writing paper except for one major concern about the fundamental methodology. Despite the topic and overall research strategy of this study is interesting and informative, the descriptions for mainly methods and results are rough and lack some information. If the authors fully explain the concerns, and the main results and discussions don’t change after the revisions, this manuscript can be acceptable.

One main concern is the analyzing position of statolith trace element. Whether the sampling point of trace elements is set in accordance with the absolute (employed in this study) or relative position form the core is the reconsidering point.

There are several places especially in the Materials and Methods and Results that lack enough explanations about each topic.

There are several gaps between the interpretation in the sentence of Results and the actual data on each figure and table.

In the Discussion, it is better to add the comparison between the data in this study and the previous studies about cephalopods and/or fishes.

The specific comments are follows.

Introduction

Line 36: Cephalopods are important not only as predators but also as preys in the oceanic ecosystem.

Line 38: Meaning of the word ‘complex population structure’ is a bit unclear.

Line 45−66: It is better to switch the order of paragraphs between Line 45−57 and 58−66 for smooth reading.

Line 78: Does the ‘small pelagics’ mean pelagic fish?

Line 81−85: The spring stock doesn’t be mentioned. Here needs to cite.

Line 85−86: Does the sentence mean that the autumn stock shows the highest growth rates among the stocks?

Line 86−89: It is hard to understand the meaning of this sentence.

Line 91: Meaning of the word ‘biological characteristics’ is a bit unclear.

Materials and Methods

Line 103−104: The information showing the area of sampling site looks different from the distribution of the dots on the Figure 1.

Line 122−123: It would be hard to understand the exact meaning what the sentence shows.

Line 126−127: The data about body weight and gonad maturity grade doesn’t be shown.

Line 131−130: About the "one-day" growth rule, it needs to cite.

Line 135−138: It needs to explain more carefully about the details of statoliths selection.

What kind of scientific meanings do the selected range for the hatching period, the mantle length distribution, and sex composition have? Why didn’t the spring stock be used?

The information showing the number of statoliths analyzed looks different from the data about the ‘Quantities’ in the Table 1.

The ‘Sex ratios’ in the Table 1 needs to show whether male:female or female:male.

Line 153−154:

There would be one potential need to reconsider whether the sampling point of statolith trace elements is set in accordance with the absolute (employed in this study) or relative position form the core. Since the growth speed of statolith, namely the length between two increment lines, should be different among individuals, the absolute position from the core of statolith should record the information of different time point among each statolith.

Even if the authors keep using the absolute position, it should be better to check the information of time point that each laser ablation circle (40 μm diameter) contained by counting the number of increment line from the core. Depending on this information, the data of trace elements should be corrected and compared. The information of time that each measurement point contained should help to make the clustering results induced by the multivariate regression tree model more reliable.

Line 162−165: It is better to consider the possibility that the absolute position from the core should contain the information of different time point among each statolith.

Line 188−189: To show validity about the assumption of unchanged relationship, it needs to cite.

Line 196: What is the reason about setting ‘one week’?

Line 197−198: About the ‘maximum swimming speed (20 km/d)’, it needs to cite.

Line 200: ‘with’ or ‘within’?

Results

Line 221: There are some ‘otolith’ in this section instead of ‘statolith’.

Line 229−231: There is no mention about ‘Fe’. It needs to explain about the data of Fe and the reason why the data didn’t be used for later analyses.

Line 232−247:

It needs to explain more carefully about the details of results possibly with showing the statistical evidence.

For example…

The Mg:Ca value of summer stock at 360 μm looks lower than the autumn value, and it is impossible to compare the data between summer and autumn at 540 μm because of no autumn data.

It appears to be double peak for the Sr:Ca value for summer and winter stocks.

The Sr:Ca value of autumn stock at 180 μm looks lower than the summer value.

The Sr:Ca value of autumn stock at 300 μm looks higher than at least the winter value.

The Ba:Ca values of all stocks looks showing from constant trend to upward trend.

The Ba:Ca value of autumn stock at 240 μm looks lower than at least the summer value.

Line 250−255: It needs to explain more carefully about the details of ‘significant differences’ among stocks.

Line 257−265: Are there any results of comparison between the statolith nodes and the age information estimated by counting the number of increment line from the core? Through this comparison, the validity of cluster results estimated by the MRT should be empirically confirmed. At least, since there is the data about the average period of embryonic stage for this species in the previous studies, it is better to mention the comparison between the age information and the first node.

The name ‘larval stage’ is not correct for the cephalopods. ‘Paralava’ is more suitable.

Line 269−279:

Does the dashed line in the Figure 5 show the four nodes estimated by the MRT? It looks the gaps between the values of length and the position of the dashed line on the x axis.

It also needs to explain more carefully about the details of results possibly with showing the statistical evidence.

Line 278−279: The result that the Sr:Ca value of summer stock in the adult stage looks increasing comparing to autumn stock is not ignorable. Between summer and winter, the trend of Sr:Ca looks similar.

Line 291−294: Table 4 doesn’t contain the data about Temp_25 and Na/Ca.

Line 290−298: There are no explanation about the results that e.g. each temperature and Mg:Ca and Ba:Ca for winter stock, some temperatures and some elements for both summer and autumn stocks. It is better to put in the sentence and/or Table 4.

Table 4 doesn’t need to mark some data with underlines.

Line 307−309: Is this integration between summer and autumn stock valid?

Line 312: There is a gap between the value of temperature at the embryonic stage in the sentence and Table 5.

Line 319−320: There is no mention about the movement to the south between the embryonic stage and the larval (paralava) stage.

Line 327−332: Figure 6 and 7 needs the explanation about the heat bar.

Line 333−335: It is better to put the labels of S1−5 on each center of distribution.

Discussion

Line 337−359: This part of Discussion looks like Introduction. To discuss the ‘Analysis of trace elements’, instead of the current story, it is better to establish a story about showing the validity of analysis methods employed in this study comparing to other methodologies about the statolith trace elements other previous studies employed.

Line 338: The data about Fe didn’t be shown and analyzed.

Line 348: It is better to write a few examples about the ‘physiological properties of individuals’. 

Line 360−401: This parts need to include the discussion of comparison between the results from this study and other cephalopods/fishes. Some studies of cephalopods were already included in the introduction.

Line 367−369: This description is incoherent with the result in Line 233−234.

Line 370−371: It is hard to understand the meaning of this sentence.

Line 375−376: This description is incoherent with the result in Line 244−245.

Line 392−393: It is hard to understand the meaning of this sentence. For the winter stock, the cold water in the early stage should increase the Sr:Ca value.

Line 402−466: This parts also need to include the discussion of comparison between the results from this study and other cephalopods/fishes. Some other representative (maybe commercial) species in the South China Sea would be useful to compare.

It is better to put the information about the monsoon and the each related oceanic current on the Figure 8 or newly drawn schematic figure to support understanding the discussion.

Line 404−405: This sentence shows a bit different image from the general image of cephalopods life that the sexual maturity emerges in later part of the life because the spawning events is at the end of life. It means that ‘the active swimming ability’ would be already employed before the maturing process.

Line 452: ‘27.15–27.92°C’ doesn’t show the correct value from the Table 5. ‘Table 6’ is wrong.

Line 457−460: The water temperature of the summer-autumn stock should keep degreasing in the Table 5 instead of the change from decreasing to increasing.

Line 460−464: It is better to discuss the relationships between the life period of each stock and the age information estimated from the number of increment line of statolith.

Round 2

Reviewer 4 Report

 Sucessfully improved

Spell check recommended